# Stratified Data Integration

Fausto Giunchiglia[0000−0002−5903−6150], Alessio Zamboni[0000−0002−4435−1748],
Mayukh Bagchi[0000−0002−2946−5018], and Simone Bocca[0000−0002−5951−4589]

Department of Information Engineering and Computer Science (DISI),
University of Trento, Italy
{fausto.giunchiglia,alessio.zamboni,mayukh.bagchi,simone.bocca}@unitn.it

**Abstract.** We propose a novel approach to the problem of semantic heterogeneity where data are organized into a set of *stratified* and *independent* representation layers, namely: *conceptual* (where a set of unique alinguistic identifiers are connected inside a graph codifying their meaning), *language* (where sets of synonyms, possibly from multiple languages, annotate concepts), *knowledge* (in the form of a graph where nodes are entity types and links are properties), and *data* (in the form of a graph of entities populating the previous knowledge graph). This allows us to state the problem of semantic heterogeneity as a problem of *Representation Diversity* where the different types of heterogeneity, *viz. Conceptual*, *Language*, *Knowledge*, and *Data*, are uniformly dealt within each single layer, independently from the others. In this paper we describe the proposed stratified representation of data and the process by which data are first transformed into the target representation, then suitably integrated and then, finally, presented to the user in her preferred format. The proposed framework has been evaluated in various pilot case studies and in a number of industrial data integration problems.

**Keywords:** Semantic Heterogeneity · Knowledge Graph Construction · Stratified Data Integration

## 1 Introduction

*Semantic Heterogeneity* [1,2], namely the existence of *variance in the representation* of the same real-world phenomenon, has long been a major impediment for effective large scale *data integration* implementations. It is inescapable in nature and is rooted in the *diversity* which is inherent in different means of representation (which itself is rooted in world diversity) [3]. The pervasiveness of semantic heterogeneity in its various forms, has long been an overarching concern in the data management research landscape [1,2]. In particular, its obtrusive ramifications with reference to *data integration* scenarios have been widely acknowledged, and *partial approaches* towards their resolution at the schema and data level have been proposed, see for instance [1,4,5,6].

In this paper we propose a novel approach which is based on the key assumption that data should be represented as a set of *stratified* and *independent* representation layers, namely:

1. *conceptual*, where concepts, codified as a set of unique alinguistic identifiers, are connected inside a graph codifying their meaning;
2. *language*, where sets of synonyms, i.e., synsets [7], possibly from multiple languages, annotate concepts;
3. *knowledge*, in the form of a graph where nodes are entity types and links are properties; and
4. *data*, in the form of a graph of entities populating the previous knowledge graph.

The representation of data according to these four layers allows us to state the problem of semantic heterogeneity as a problem of *Representation Diversity*, where heterogeneity distributes itself over these layers thus being stratified into four different types of diversity, *viz. Conceptual*, *Language*, *Knowledge*, and *Data*, which can then be are uniformly dealt within each a single layer, independently from the others. The proposed approach has three major advantages. The first is that the combinatorial explosion deriving from the interaction of the four different types of diversity is avoided and the complexity of the data integration problem reduces to the sum of the complexity of each layer. Each layer can be dealt with *as if* all the other layers presented no heterogeneity at all. The second is that the techniques developed for each layer can be composed with the ones developed in the other layers irrespectively of how heterogeneity appears in the current problem. The third, which is a direct consequence of the second, is that, within each layer, it is possible to exploit the large body of work which can be found in the literature (see the related work section for a detailed discussion on this point).

In this paper, after restating the semantic heterogeneity problem as a representation diversity problem, we describe the proposed stratified representation of data and the process by which data are first transformed into the target representation, then suitably integrated and then, finally, presented to the user in her preferred format. The main contributions of this paper can be articulated as follows:-

1. a novel articulation of the problem of semantic heterogeneity as stratified into the above four layers of representation diversity (Section 2);
2. a viable solution to the issues above in the form of an *end-to-end* logical data management architecture grounded in our four layered stratification of representation diversity, wherein we focus on accommodating the heterogeneity resident in each layer, independently from all the other layers (Section 3);
3. an illustration of an implemented semi-automated data integration pipeline which exploits the representation of diversity presented in Section 3 (Section 4).

The pipeline presented in Section 4 has been extensively evaluated within many data integration pilot studies, which have spanned three years and has then applied in various real world problems. Section 5 presents a short highlight of these activities. Towards the end of the paper, Section 6 contextualizes the contribution of our work by surveying the state of the art, while Section 7 outlines the future research ventures.

Table 1: Three semantically heterogeneous schemas containing a heterogeneous description of the same entity.

| Car | | | | |
|---|---|---|---|---|
| **Nameplate** | **schema: speed** | **schema: fuelCapacity** | **schema: fuelType** | **schema: modelDate** |
| FP372MK | 150 | 62 | Petrol | 2020-11-25 |

| Vettura | | |
|---|---|---|
| **Targa** | **Velocità** | **Tipo di corpo** |
| FP372MK | 158 | Coupé |

| Vehicle | | | |
|---|---|---|---|
| **vso:VIN** | **vso:feature** | **vso:modelDate** | **vso:speed** |
| FP372MK | Armrest | 2020-11-25 | 155.0 |

## 2    The Stratification of Diversity

We show how semantic heterogeneity can be stratified in four independent diversity problems via the following motivating example ( see also Table 1).

*There are three datasets {Car, Vettura, Vehicle} which refer to the same real world entity, namely a car, which we assume has plate 'FP372MK'. The first dataset describes FP372MK as a car, having five attributes, four of which are expressed using the automotive extension of schema.org.[1] The second dataset also considers the entity as a car but its description is provided in Italian. The third dataset encodes the entity as a vehicle having four attributes expressed employing the Vehicle Sales Ontology[2] namespace 'vso:'.*

By a close look at the above example, it is easy to notice four different types of diversity which we can briefly describe as follows:

- *Conceptual diversity (L1):*[3] the same real world object is mentioned in two datasets using the concept denoted by the word car while in the third data set is called vehicle, namely using a more general term (because, for instance, in this latter case there is no need to distinguish among the various types of vehicles as the issue is that of counting the number of free parking lots).

---

[1] https://schema.org/docs/automotive.html

[2] http://www.heppnetz.de/ontologies/vso/ns

[3] L1, L2, L3, L4, L5 are the five layers into which we organize the representation of data. L3, the layer used to represent causality [8], is not discussed here because it is irrelevant to the goals of the paper.

- *Language diversity (L2)*: the same real object is described using three different lexicons, namely, that of a natural language, i.e., Italian, and two namespaces, i.e., from the automobile extension of schema.org and from the Vehicle Sales Ontology, both of which use a different natural language, i.e., English, as base language. Notice that these are three different lexicons, independently developed where, therefore, the meanings of the terms used is intuitively similar but formally unrelated.
- *Knowledge diversity (L4)*: the same real world object is described using different properties, the motivation being most likely in the different focus of the three databases. Thus, for instance, the first could be the description used in an online car rental which codifies its data using schema.org, the second could be the description used by the Italian Automobile Club, while the third could be the description used in an online sales portal.
- *Data diversity (L5)*: the same real world object is described in a way that, even when associated with the same properties, the corresponding values are different. There can be many reasons for this last source of heterogeneity, for instance, different approximations, different formats, different units of measure, different reference standards (e.g., date standards for dates) and so on.

Let us consider these four layers in detail.

*Conceptual Diversity.* The notion of concept is well known in the Philosophy of Language literature, see, e.g. [9], and in Computational Linguistics, see, e.g. [7].. Here we follow our own work, as described in [10,11], and take concepts to be *unique alinguistic identifiers*. Concepts are organized in multiple hierarchies, one per syntactic type (i.e., noun, verb, adjective) wherein a child (father) concept is taken to be more specialized (more general) than the father (child) [7,11]. For instance noun hierarchies are organized in terms of *'hypernym-hyponym'* links where, in the example in Table 1, the term car is a direct hyponym of the term vehicle.

*Language Diversity.* Languages, taken here in a very broad sense to include, e.g., natural languages, namespaces and formal languages. Language diversity occurs both *across and within languages*. Thus, there are multiple languages available for the purpose of representing the same concept, but also, even within the same language, linguistic phenomena like *polysemy* and *synonymy* allow for multiple diverse representations of entities. As a result there is a *many-to-many mapping* between words and concepts, both within the same language and across languages [10,11].

*Knowledge Diversity.* We model knowledge as a set of *entity types*, also called *etypes*, meaning by this, classes of entities with associated *properties*. Knowledge diversity arises from the *many-to-many mapping* between etypes and the properties employed to describe them [12], and can appear in one of two different forms. The first appears when there are '$n$' representations of different etypes described in terms of the same set of properties. The second manifests itself when there are '$n$' representations of the same etype with different sets of properties.

As an example of the latter situation, in Table 1 two datasets describe the same etype car, but the two etypes are associated three different groups of properties.

*Data Diversity.* We model data, meaning by this the concrete, ground knowledge, that we have about objects in the world, as *entities* each associated with *property values*, where properties are inherited from the etype of the entity. Data diversity [1] exists because of the fact that the *mapping* between entities and the property values used to describe them is *many-to-many*. Data diversity appears as well in two different forms, wherein the same real world entity, associated with the same properties, is described using different data values, while dually, two different real world entities, still associated with the same properties, can be described using the same data values. As an example of the latter situation, there can be two identical cars which are both described by a set of attributes which do not contain their plate or any other identifying attribute. The example in Table 1 provides an example of the former situation. Here, the three datasets refer to the same entity, a car with plate 'FP372MK', which shares a common attribute which is the car speed, but this property has three different values.

Notice how the stratification of diversity presented above has the following crucial characteristics. The first is that each layer models a different type of phenomenon and the corresponding type of diversity. The second is that how diversity appears in one layer is completely independent of how it appears in any other layer. The third is that L1, the conceptual layer, provides the grounding for unifying the various types of diversity as they appear in the other layers. In fact, in all respects, L1 is a *logical theory*, which can be codified in a *logical language*, e.g., Description Logics, where the semantics of all terms (the alinguistic identifiers) is univocally defined by the links of the hierarchy. The fourth and last observation is that the diversity mappings which appear in L2, L4 and L5 are all *many-to-many* and this generates the type of combinatorial complexity which makes it so difficult to handle the problem of semantic heterogeneity. As already hinted to in the introduction, the stratification of semantic heterogeneity provides a major advantage in that it allows to structure it in four independent and much smaller problems, where each problem can be treated uniformly by developing methods and techniques which are specialized just for that layer. This last observation is the basis for the work presented in the remaining of this paper.

## 3 Representing Diversity

Fig. 1 depicts the proposed *data management architecture* instantiated (partially, for lack of space) to the example in Table 1. Here the arrows represent the *functional dependencies* which must be enforced among the different layers and, therefore, implicitly define the *order of execution* which must be followed during a data integration task, starting from the user input and concluding with the fully integrated data. Fig.3 in Section 4 will later depict the process, tools and algorithms which exploit the different representation layers in Fig.1 towards producing, in the end, the target data integration.

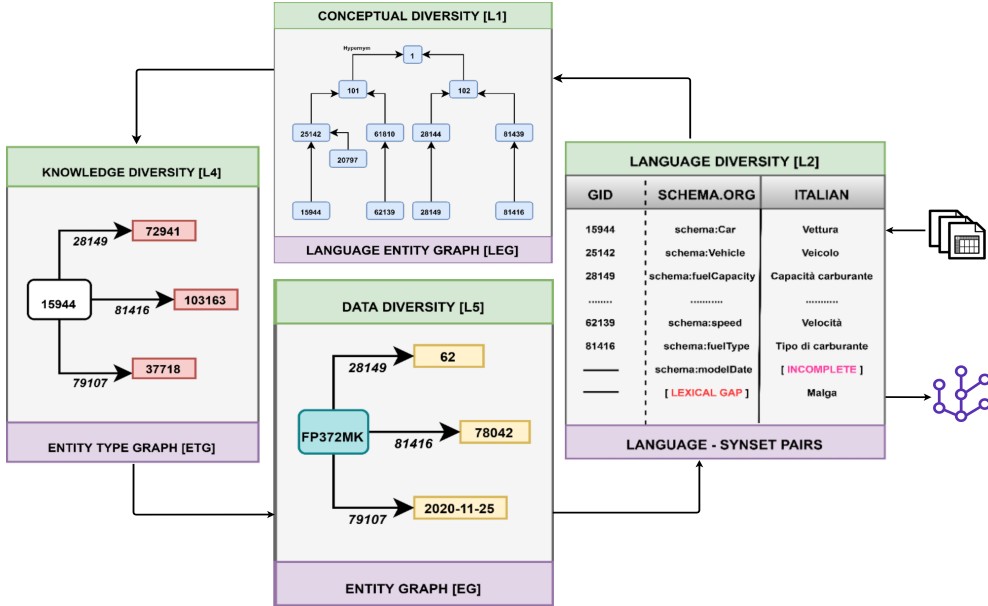

Fig. 1: The Representation Diversity Architecture

The language representation layer (L2) appears first and last in the architecture in Fig.1. L2 enforces the input and the output dependence of the representation of data on the user language. In fact, language is the key enabler of the bidirectional interaction between users and the platform. In the first phase, the L2 input language is translated into the system internal L1 conceptual language and the input language is only resumed during the last step, when the results of the data integration steps are presented back to the user. To this extent, notice how, in Fig.1 the language used in the LEG (L1), ETG (L4) and EG (L5) are just ids, while the conversion table of the first phase is the repository is the mapping from internal and external language(s). In this process, L2 is key in keeping completely distinct the multilingual user-defined data representation and the alinguistic system-level data representation. It is also important to notice how the proposed data management architecture is natively multilingual as a result of the L1 alinguistic concepts being the convergence of semantically equivalent words in different languages. A very important case which can be dealt by this architecture is the *heterogeneity of namespaces*, as also reflected in the running example. Any number of namespaces and natural languages can in fact be seamlessly integrated following the same uniform process.

The management of conceptual diversity (L1), which functionally comes next in sequence, involves the organization of the L1 alinguistic concepts, as identified in the first step, into a *Language Entity Graph (LEG)* which codifies the semantic relations across concepts (and, therefore, among, the corresponding L2 input words). In order to achieve this goal we exploit, as a-priori knowledge, a multi-

lingual lexico-semantic resource, called *UKC (Universal Knowledge Core (UKC)* [10,11] which represents words, synonyms, hyponyms and hypernyms quite similarly to the Princeton Wordnet [7], still with important differences [13]. The net result of this phase is an LEG with the following properties:

- the concepts identified during the first phase are all and only the nodes in this graph;
- these nodes are annotated with the input L2 terms, across languages;
- these nodes are organized into a hierarchy which preserves the ordering, across the links of the UKC (in the case of nouns, the synonym/ hyponym/ hypernym relations).

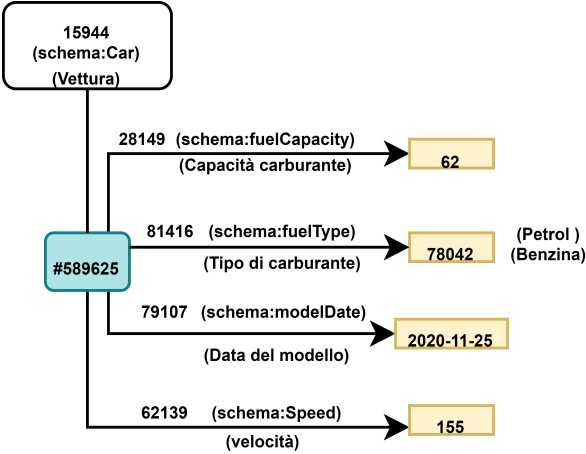

Fig. 2: An example Entity Graph (EG)

The third representation layer (L4), dedicated to the management of *knowledge diversity*, involves the construction of a (alinguistic) *Entity Type Graph (ETG)* encoded using *only* concepts occurring in the LEG costructed during the previous two phases. In this phase, the first step is to distinguish concepts into *etypes* and *properties* (both object properties and datatype properties) while the second step is to organize them into a *subsumption hierarchy*. The key observation here, which constitutes a major departure from the previous work is that *etype subsumption, as encoded in the ETG, is enforced to be coherent with the concept hierarchy encoded in the LEG*. Thus for instance, as from the above example, the object with plate FP372MK, being a car, can be encoded to be an entity of etype vehicle, but *not* of, e.g., etype organism. This fact, which is natively enforced by the lexico-semantic hierarchy of the UKC for what concerns natural languages (in the above example, Italian), is extended to cover also the terms belonging to namespaces (in the above example that of schema.org

and that of the Vehicle Sales Ontology). This alignment of meanings across languages and namespaces, which absorbs a major source of heterogeneity present in the (Semantic) Web is a natural consequence of the language and conceptual alignment performed during the first two phases.

In the fourth representation layer (L5), we tackle the heterogeneity of data values by employing an *Entity Graph (EG)*, namely, a *data-level knowledge graph* populating the *ETG* with the entities extracted from the input datasets. Fig.2 reports the EG resulting from the first four phases. As you can notice this graph is constituted of a backbone of L1 alinguistic ids, each annotated with the input L2 terms where, for each L2 term, the system remembers the dataset it comes from. This information is crucial in case of iterated (multi-phased) data integration, as it is usually the case, as the system needs to remember which new terms and values substitute which old terms and values. This mechanism is implemented via a *provenance* mechanism, not represented in Fig.1, which applies to all the input dataset elements, both at the schema and at the data level. A last observation is that in Fig.1 the unique id identifying the car with plate FP72MK is #589625, a new identifier which never appeared before. In fact, any time a new entity is identified, it is associated a unique id which is managed internally by the system and that the user can see and also query, but not modify.

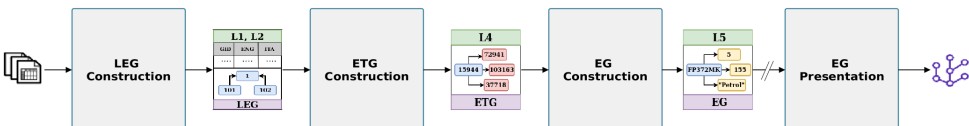

Fig. 3: The Representational Diversity Pipeline

## 4   Processing Diversity

The pipeline in Fig.3 describes the process used to manage and integrate the diversity as it appears in the four layers described in the previous section. This *partially automated* pipeline is highly *flexible*, *independent* of the input data and domain, and largely *customizable*. As detailed in the next section, it has been applied to the modeling of events, facilities, personal data, medical data [14], university data [15] and many other domains. Its intended target users are *Data Scientists* with no programming knowledge but with an understanding of the domain and data integration problem to be solved. All tasks are managed via a flexible user-friendly user interface and the process is assumed to start with the input datasets being in a repository from where they can be uploaded. The only required programming effort, which we assume should not be performed by the data scientist, is that needed to extract the input datasets from, e.g., legacy systems or open data repositories, and to pre-process them.

The pipeline is composed of four main phases, as depicted in Fig.3, where the first phase produces the integrated L1 and L2 representation of the LEG,

the second phase produces the ETG and the third phase produces the EG. The fourth phase, the phase of *EG Presentation*, depicted in Fig.3 for completeness, is representative of an external client application exploiting the LEG, ETG and EG produced by the pipeline. During this phase, not further discussed, the data scientist usually selects the language in which she wants the EG to be presented to her, e.g., using the words of the input datasets or any other language supported by the UKC.[4]

*LEG Construction*: This first activity takes in input all data to be integrated (in the example above, this consists of all the three tables represented in Table 1), and it extracts each word and multi-word occurring in the input tables (in the case of namespaces, the namespace itself and the word are treated as a single concept). The output of this phase is the L1 and L2 representation of the input data organized in a LEG. During this phase the prior knowledge codified in the UKC is heavily exploited (see the previous section for details). This activity is performed with the help of the Word Sense Disambiguation (WSD) component SCROLL, a multilingual NLP library and pipeline which is specialised to handle the *Language of Data*, as defined in [13], namely the type of NLP sentences that are usually found in data. At the moment SCROLL supports seven languages (including various European languages but also Mongolian and Chinese) but, as we have found out, because of the similarity of many languages, of the relatively simple structure of the language of data, and of the fact that the system processing is in control of the Data Scientist which validates each step, SCROLL can also be useful in various other similar languages (where similar here means *not diverse*, with language diversity being defined as in [11]). The main features of SCROLL which make it quite suitable for multilingual data integration are:

- it has been developed to be highly modular and with a clear split between the modules which are language independent from those which are language dependent;
- in SCROLL, the tasks that are language dependent and must therefore be implemented for each new language (e.g., word segmentation in English and Italian is very different from word segmentation in Chinese) are executed as soon as possible. The net advantage is that the more semantics dependent tasks, e.g., Entity Recognition (ER) and Word Sense Disambiguation (WSD) work on a conceptual representation of the data and are therefore implemented once for all;
- SCROLL's NLP pipeline is highly optimized and fully integrated with the UKC, mainly with the goal of implementing a highly optimized and highly effective language agnostic WSD task which is also domain aware [16];
- it is often the case SCROLL encounters new words which are not in the UKC which, in turn may or may not contain the corresponding concept id. These situations are dealt with by suitably enriching the UKC according to a dedicated mechanism.

---

[4] In the current state of implementation, this phase can only perform a word by word translation without being able to reconstruct the overall meaning of a sequence of words.

The output of this phase is a spreadsheet, including the structured definitions of new concepts and their relations, which, suitably interpreted on the basis of the UKC hierarchy, codifies the LEG.

*ETG Construction*: This activity takes in input the schemas of the input datasets, where all the words are now annotated with the LEG concepts and it constructs the ETG which integrates them. This phase is performed via a *Knowledge Editor*, similar in spirit to Protègè,[5] but highly integrated in the pipeline in Fig.3, which is used interactively by the data scientist. Two are the main operations which are performed during this phase with the help of the knowledge editor:

– perform schema matching. This activity is performed manually but it benefits from the suggestions provided by a multilingual schema matcher [17] (where, however, an effective way to integrate this matcher in the knowledge editor is yet to be found);
– build the resulting integrated ETG and possibly align it with a reference ontology. The goal of this step is to produce a clean and highly reusable ETG. This step at the moment is performed manually, but the plan is to integrate this functionality inside the Knowledge Editor, exploiting the results described in [12].

The output of this phase is an OWL file codifying the ETG where all the terms which are used are annotated with the LEG's concept ids.

*EG Construction*: This activity takes the ETG and the input datasets, annotated with the LEG concepts, and constructs a mapping between the data values within each dataset and the ETG built during the previous step. The EG Construction is an iterative activity which considers one dataset at the time. The mapping operations are implemented through the usage of a specific tool, called *KarmaLinker*, which consists of the integration of the *Karma* data integration tool [4,5], which does not do any NLP, with SCROLL. Within each activity iteration, KarmaLinker maps the data values in the input dataset to the etypes and properties of the ETG. Some of the most important and non trivial operations performed here are that of recognizing the entities which are implicitly mentioned in the input datasets (*entity detection*), of recognizing their entity types (*etype recognition*) and, finally, of recognizing whether there are multiple occurrences of the same real world entities, possibly described using different properties and property values (*entity mapping*, see the example of data heterogeneity presented in Section 2). In the example of Fig.1, all the three input datasets are recognized to describe the same car which, as from Fig.2, is then assigned the unique id #589625. The final output is an EG encoding the information in the initial datasets, at all the four different levels of diversity, stored using a language agnostic JSON-LD [6] format. See Fig.2 for a partial representation of the EG constructed from the datasets in Table.1.

---

[5] https://protege.stanford.edu/
[6] https://json-ld.org/

## 5    Evaluation

The representation architecture and pipeline described above, have been experimented and evaluated during the past three years (2018, 2019, 2020), the last year still being ongoing, as part of the Knowledge and Data Integration (KDI) class, a six credit course of the Master Degree in Computer Science of the University of Trento.[7] Table 2 reports the information regarding the population involved (excluding PhD students which are not counted) and number of projects. During this class students, 2-5 people per group, must generate an EG using the pipeline above starting from a high level problem specification. Part of the task is also to identify the most suitable datasets and pre-process them. Datasets are usually found in `open data` repositories but some of them are also scraped from the Web. The overall project has an elapsed time of 14 weeks during which students have to work intensely, even not full time. We estimate the overall effort that each group puts into building an EG in around 4-8 man-months, depending on the case. At the end of the course, after the final exam, students are asked to evaluate the methodology they have used (as partially described in this paper). This evaluation involves various aspects including application scenarios, datasets used, ETG and EG generation, language management and LEG generation, and evaluation of the overall pipeline.[8] As of to day we have piloted 24 projects and 75 evaluations.

Table 2: Evaluation's subjects in KDI class - 2018, 2019, 2020.

|                  | 2018 | 2019 | 2020 | Tot |
|------------------|------|------|------|-----|
| # students       | 29   | 20   | 26   | 75  |
| # project teams  | 14   | 4    | 6    | 24  |
| % Male           | 69%  | 75%  | 95%  | 59  |
| % Female         | 31%  | 25%  | 5%   | 16  |

A first question in the evaluation is about L2 and the management of language diversity, with a specific emphasis on the use of the UKC. The percentage of students which believe that the explicit management of language diversity, as a dedicated independent phase, is worthwhile was 79.4% in 2018 and 80% in 2019. A second question is about L4 and the management of knowledge diversity. More specifically, here the purpose of the question is to understand if students find

---

[7] See https://unitn-kdi-2020.github.io/unitn-kdi-2020/ for more details. This site contains the material used during the 2020 edition of the course and it consists of theoretical and practical lectures, as well as demos of the tools to be used, some of which have been mentioned above.

[8] We report below only the statistics for the first two years. This year's statistics, which will be available soon, will be added in the final version of the paper.

it helpful to define the etypes of the input data, and if the pipeline properly supports this task. The 69% of the students in 2018, and 95% in 2019, provided a positive answers. Moreover the 72.4% (2018), and 60% (2019), stated that grounding the types with the reference ontology, usually an upper ontology, facilitates the construction of the ETG and in particular the positioning of the entity types. A last set of questions are asked about the *overall usability* of the methodology, where each question aims at the evaluation of a specific usability aspect. The answers are reported in Fig.4 for both 2018 and 2019, over a 1 to 7 scale (1 maximally positive, 7 maximally negative). Observing the figures below we can notice an overall positive trend. Furthermore we can see that, with respect to 2018, in 2019 students found the methodology easier to use and also easier of learn. Moreover, we noticed that the level of efficiency in the accomplishment of the data integration objectives increased in 2019. This is part of an overall positive trend that, we believe, will also be confirmed in 2020 and that, we believe, relates to the continuous adjustments to the methodology and to the tools that we perform every year, also based on the feedback collected during the KDI course.

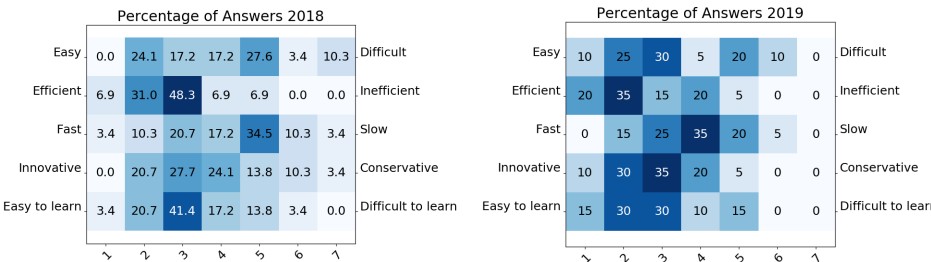

Fig. 4: Usability of the methodology: (a) 2018, (b) 2019

## 6 Related Work

As from the introduction, our approach to representing and managing semantic heterogeneity as the stratification of four independent problems, was never proposed before. However, the very same fact that we are stratifying semantic heterogeneity into the four problems of conceptual diversity, language diversity, knowledge diversity and data diversity, one at the time, allows us to refer and exploit the huge amount of work which has been independently done in these areas. The following of this section concentrates on this work, across the four layers, including also earlier work from the authors.

The notion of language diversity, as described here, was first introduced in [10,11] which also provide a detailed description of the UKC. However, this work builds upon decades of the work in the development of *multilingual lexical resources*, see, e.g. [7,18,19]. The main innovation with respect with this earlier

work is that LEGs, and the UKC in particular, have a separated and independent conceptual layer while, in the previous lexico-semantic resources, L1 and L2 are collapsed. The stratification between L1 and L2 on one side and L4 on the other side, and the exploitation of lexical resources in order to do schema integration is a direct application of the ideas and technology developed in the field of ontology matching, see, e.g., [20,21,22,23]. The work in [17], used during the ETG construction, builds upon this previous work by proposing *NuSM*- a multilingual ontology matching framework which heavily exploits the UKC.

Our proposal of using knowledge graphs is very much in sync with the huge amount of work now being developed in this area. Differently from the previous work, we uniquely stratify Knowledge Graphs into four layers, however, see [24,25,26,27] for an approach which is quite similar in spirit to ours, in particular in the distinction between L4 and L5. Furthermore, the validity of an ontology guided, knowledge graph backed approach towards data integration and presentation has been favourably discussed in [28,29,30].

In the context of *semantic data integration* [31,32], the survey in [33] noted the prevailing difficulties as non-standardized identity management, multilinguality management, data and schema heterogeneity, namely all issues which our work addresses. The work in [6] also mentions architectural, structural, syntactic and semantic heterogeneity in data integration frameworks, all issues that our proposed approach tackles. The work in [34] and [35] combined, highlights diverse *parametric* aspects of six major openly available knowledge graphs, with [34] calling for newer approaches in knowledge modeling and new forms of knowledge graphs. More specifically, Wikidata [36] emerges as a feature-rich, cross-domain openly available knowledge graph [34]. Still, due to its very nature, with respect to the work proposed here, the work on Wikidata lacks an adaptive schema customizable to different data integration scenarios, and an overall explicit, stratified data management architecture.

## 7    Conclusion

In this paper we have presented an innovative organization of data management stratified across four layers of heterogeneity - namely concept, language, knowledge and data. This has allowed the re-interpretation of semantic heterogeneity as a problem of representation diversity and the proposal of a stratified logical architecture which deals with this problem. The future work will consist of a generalization of the pipeline presented in this paper into a full-fledged knowledge graph based methodology for data integration.

## Acknowledgements

The research conducted by Fausto Giunchiglia, Mayukh Bagchi and Simone Bocca has received funding from the *"DELPhi - DiscovEring Life Patterns"* project funded by the MIUR Progetti di Ricerca di Rilevante Interesse Nazionale (PRIN) 2017 – DD n. 1062 del 31.05.2019. The research conducted by Alessio

Zamboni was supported by the *InteropEHRate* project, co-funded by the European Union (EU) Horizon 2020 programme under grant number 826106.

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
