# OpenReview forum: "Stratified Data Integration"
_eswc-conferences.org/ESWC/2021/Workshop/KGCW — KGCW 2021_

### Official Review · ~Herminio_García-González1 · 2021-04-01
**Interesting proposal that loses in the details**

**Rating:** 6
**Confidence:** 3

**Review:**

This article presents a proposal for data integration based on language technologies. It divides the data integration task in four different specialised layers which, involving domain experts in the process, through different phases can reach a semi-automatic integration in the form of a knowledge graph.

I find the idea in this paper very interesting as it offers a different perspective on data integration tasks. Namely, the use of different layers seems like a very good approach which could evolve in more complex and accurate systems in the future. However I have some concerns about some paper aspects.

First of all is quite surprising to have the Related Work section just before the conclusions. Although it could be placed here I think that is not convenient in the case of this paper. I think that related work should position this work among the rest of semi-automatic data integration approaches, then show how it differs from them and finally explain the whole solution. In addition, current related work section builds in used language technologies and barely nothing in data integration. When reading this section some works come to my mind like [1-2] which are not completely equal but it would be worthy to compare to. Besides, it also builds a lot in Knowledge Graph literature as a sort of motivation. Again, related work should be used to compare to the state-of-the-art techniques, not as a motivational part that should be included in the introduction.

It is said that programming efforts are needed to pre-process and clean data from legacy datasources or open-data sources. Couldn't this spoil all the benefit from your system? You are proposing a system for domain experts which easies the semantic data integration but for certain legacy datasets (a lot of data indeed) you need a developer to pre-process and clean them which make me think if it is not more straightforward for the developer to perform directly the mapping using some declarative mapping language. Results for fast-slow measure in Fig. 4 also drive me to think that.

Looking to the evaluation section, you estimate the building of an EG in 4-8 man-months. I always find this man-months unit quite vague as every person is a whole different world. I would rather go for n months for n people in a group. In footnote 7, you give a lot of detail through your course webpage which is nice for transparency. However, I would also find useful to have a webpage/repository for the material concerning this paper. It would be also appropriate to have here raw datasets with students' responses to favour paper reproducibility. In Fig. 4 you used a 7 point scale, why to use this and not a 5 point one? Two ideas related to the results in Fig 4.: (1) to merge all results and analyse all of them jointly to have a higher sample size, (2) to test for significant differences between years and explain possible differences with a changelog of your in-between years updates. In Table 2, in 2018 you had 14 teams with 29 students but in 2019 you had only 4 teams with only 9 students less. Why this change of criteria? Could it be a way to improve overall results in 2019 and 2020? If so, it would be nice to have a little sentence on that regard.

Another idea that could boost your evaluation part is to go for a mixed-method approach. Right now, you are only relying in a subjective analysis of students' perception. I would suggest to also include some objective measure into the analysis (e.g., final scores obtained in the course).

In the end of page 9 it is said: "enriching UKC to a dedicated mechanism". Please, specify a bit this mechanism, now it is too abstract.

Just out of curiosity, how the general procedure would work if in Table 1 we have an IRI instead of a literal value? Would it apply the procedure recursively? And what about the mentioned conciliation like in speed?

In conclusion section, it would be better to name it conclusion & future work (more accurate). Also the last sentence about future work seems so vague, without any detail. I would expect one or two phrases more about the envisaged methodology.

In Section 3, you say "(partially, for lack of space)". I would suggest to include a link to complete Figure as a supplementary material, this could improve overall understandability of the algorithm.

Some typos per section:

General:

Fig.n -> Fig. n

Abstract:

to the user in her preferred format -> in the user's preferred format (favour gender-neutral writing)

Introduction:

which can then be are uniformly dealt within each a single layer (needs some rephrasing)

The Stratification of Diversity:

the meanings of the terms used is intuitively similar -> the meaning of the terms used are intuitively similar

Computational Linguistics, see, e.g., [7].. -> Computational Linguistics [7]. OR Computational Linguistics (e.g., [7]). (keep it simple, and avoid double introductions like see and e.g., check for more along the text)

but the two etypes are associated three different groups -> but the two etypes are associated to three different groups

while the conversion table of the first phase is the repository is the mapping -> while the conversion table of the first phase in the repository is the mapping

Representing Diversity:

called UKC (Universal Knowledge Core (UKC) [10,11] -> called Universal Knowledge Core (UKC) [10,11]

of a (alinguistic) Entity -> of an (alinguistic) Entity

in the LEG costructed -> in the LEG constructed

Processing Diversity:

the input datasets being in a repository from where they can be uploaded (rephrase)

takes in input all data -> takes as input all data

Table.1 -> Table 1

Evaluation:

provided a positive answers -> provided positive answers OR provided a positive answer

easier of learn -> easier to learn


[1] Futia, G., Vetro, A., Melandri, A., & De Martin, J. C. (2018). Training Neural Language Models with SPARQL queries for Semi-Automatic Semantic Mapping. Procedia Computer Science, 137, 187-198.

[2] Futia, G., Vetrò, A., & De Martin, J. C. (2020). SeMi: A SEmantic Modeling machIne to build Knowledge Graphs with graph neural networks. SoftwareX, 12, 100516.

---

### Official Review · ~Hannes_Voigt1 · 2021-04-12
**The paper presents an approach to data integration that deals with different kinds of diversity step-by-step.**

**Rating:** 6
**Confidence:** 3

**Review:**

Dealing with the different kinds of diversities separately appears (a) a fairly obvious approach and (b) not a new thing as the existence of the specialized tools the authors use suggest. That is to say, the separation of the integration problem into a subproblem is not original.

The higher-order bits of the presented approach, i.e. which subproblem to tackle in which order and how to integrate specialized tools for the individual subproblem into a consistent and seamlessly useable process may very well be original. The main benefit of the presented approach appears to be first and foremost in this integration of data integration tools in one process.

The major weak point is the evaluation:

While the evaluation appears to show that the approach allows students to solve that assignments and makes the think they did it in a relatively efficient way, a couple of question remain unclear:

* Have student evaluated other tools/approaches in comparison?
* Do the student even know other tools/approaches?
* In which degree is the selection of datasets biased by the knowledge about and the capability of the used tool/approaches? In a real data integration problem, the to be integrated data source are likely given.
* How much time have the students actually spent on the data integration work with the tool/approaches? 4-8 man-months for completing data integration task does not sound like what makes data scientists very productive.

It would have been helpful to mention the target user group of the presented approach early on, e.g. in the introduction of the paper. As it is, it appears on the first read until like page 9 as if the paper tries to tackle the inescapable problem of diversity in data representation by introducing yet another representation.

---

### Official Review · ~Francois_Scharffe1 · 2021-04-13
**This paper presents an architecture and a system for integrating heterogeneous data sets.**

**Rating:** 6
**Confidence:** 4

**Review:**

This paper presents an architecture and a system for integrating heterogeneous data sets. The input format of data isn't precised in the paper but it is safe to assume RDF datasets are supported. The paper mentions "pre processing" as one of the tasks the user needs to perform.
The system is introduced based on an architecture or framework divinding the integration task into a set of layers that each represent an abstraction layer of the data sets to be integrated. All data sets are integrated at each layer.
The hypothesis presented in the paper is that the separation of tasks using these levels of abstraction will facilitate the integration task, while allowing to reuse the body of work in the literature available to deal with issues at each layer.
The layers are as follows:
- a conceptual layer that represents abstract terms (named alinguistic terms in the paper) represented by an id that I understand is extracted from an external thesurus. Abstract terms are organized in this layer in a is-a hierarchy. This effectively forms a sort of abstract taxonomy and allows datasets using different but related terms to be reconciled on the same abstract term (eg in the case of synonyms or multi lingual data set).
The caveat there is that if the term is not present in the external thesauri it becomes a manual task to add the terms to the thesauri.
- a language layer serves as an index between terms in the source datasets and the identifiers coming from the external thesauri.
the
- a knowledge layer that contains concepts and properties and corresponds to an ontology. Matching each dataset or schema is performed through a user interface. the output is also an ontology in OWL.
- a data layer that instanciates the knowledge layer with data from the integrated data sets. This part seems mostly automated in the implementation.
The tool and associated methodology are evaluated by a set of Master students. The evaluation is presented compared over 2-3 years depending on the parts (current year is ongoing). The evaluation is as much an evaluation of the system as it is an evaluation of the evolution of the system across these years.

The approach is original. The presentation of the architecture/framework is sometimes confusing as it is originally presented as a set of layers, then illustrated as a cycle. There are many terms introduced and the layer notation L1-L5 is then replaced with acronyms representing layers output.
The evaluation implicitly states that the system implementation seems to have evolved over the years. The implementation makes use of a number of tools into what resorts as a complex and heterogeneous architecture.
The paper would not convince me particularly to use the system as I would be wary about its complexity in having to grasp multiple tools and a significant amount of manual work. It is however a reference implementation of an archtitecture and should probably be considered as such rather than as a production ready system.

What often misses in semantic data integration tools and systems is a comparison of the time taken to do the task using the tool, versus implementing an adhoc transformation system.
As an additional note giving the task of performing data and schema matching and integration to a data scientist as stated in the paper is most probably not the best use of their time. A domain expert would probably be most efficient for the schema matching part. Maybe a job for the newly introduced knowledge scientist?

---

### Official Review · ~Maria-Esther_Vidal2 · 2021-04-20
**The paper tackles the problem of semantic data integration in the context of knowledge graph and presents an architecture to solve interoperability issues while data is integrated.**

**Rating:** 8
**Confidence:** 5

**Review:**

The paper tackles the problem of semantic data integration in the context of knowledge graph and presents an architecture to solve interoperability issues while data is integrated.
The architecture follows a pipeline were heterogeneous multilingual data present in different formats (e.g., unstructured and structured) is into an entity graph. Unique identifiers are created, allowing thus the creation of holistic profiles of the instances of each entity type.  The proposed architecture has been utilized in a Master course of Knowledge and Data Integration and evaluated by the students. The evaluation was done in terms of the type of interoperability issues that the architecture enables to solve. The results are positive and show an improvement over the time.

Overall the paper is well written and presents solid architecture which has been used to develop academic projects. This makes the results of this work worth it to be presented at the workshop.

Despite the positive points of the reported work, the current version of the paper lacks of a deep analysis of the performance of the proposed architecture. Moreover, nothing is mentioned about this architecture compares to reference architectures for data spaces (e.g., the International Data Spaces [1]) and different W3C standards for mapping representation (e.g., R2RML), integrity constraints (e.g., SHACL), and query processing (e.g., SPARQL).  Lastly, several related works are not discussed (see the list of references below).

The recommendation is for acceptance. However, the authors should discuss how the use of existing standards and reference architectures will benefit the proposed approach.


[1] Sebastian R. Bader et al., The International Data Spaces Information Model - An Ontology for Sovereign Exchange of Digital Content. ISWC 2020

Enrique Iglesias, Samaneh Jozashoori, David Chaves-Fraga, Diego Collarana, Maria-Esther Vidal: SDM-RDFizer: An RML Interpreter for the Efficient Creation of RDF Knowledge Graphs. CIKM 2020

Samaneh Jozashoori, David Chaves-Fraga, Enrique Iglesias, Maria-Esther Vidal, Óscar Corcho:
FunMap: Efficient Execution of Functional Mappings for Knowledge Graph Creation. ISWC (1) 2020

---

### Meta-Review · Program_Chairs · 2021-04-20

**Recommendation:** Accept
**Confidence:** 5

**Metareview:**

This paper presents a novel approach to address the problem of semantic heterogeneity based on a data management architecture grounded in a four layered stratification of representation diversity. All reviewers agree that the paper is worth being presented at the workshop and expressed very positive comments about the innovative aspects of the proposed approach. However, all reviewers mentioned as well that the papers lacks of an extensive comparison with the state of the art either in terms of related work or comparative evaluation. Thus, we would suggest to the authors to include a more extensive related work section where at least the relevant suggestions from the reviewers are included for the camera ready version.

---

### Decision · Program_Chairs · 2021-04-23

Accept